# Chromosome End Diversification in Sciarid Flies

**DOI:** 10.3390/cells9112425

**Published:** 2020-11-05

**Authors:** Eduardo Gorab

**Affiliations:** Departamento de Genética e Biologia Evolutiva, Instituto de Biociências, Universidade de São Paulo, Rua do Matão 277, Cidade Universitária, São Paulo 05508-090, Brazil; egorab@usp.br; Tel.: +55-11-3091-80-61

**Keywords:** diptera, repetitive DNA, *Rhynchosciara*, Sciaridae, subtelomeres, telomeres, *Trichomegalosphys*

## Abstract

Background: Dipterans exhibit a remarkable diversity of chromosome end structures in contrast to the conserved system defined by telomerase and short repeats. Within dipteran families, structure of chromosome termini is usually conserved within genera. With the aim to assess whether or not the evolutionary distance between genera implies chromosome end diversification, this report exploits two representatives of Sciaridae, *Rhynchosciara americana*, and *Trichomegalosphys*
*pubescens*. Methods: Probes and plasmid microlibraries obtained by chromosome end microdissection, in situ hybridization, cloning, and sequencing are among the methodological approaches employed in this work. Results: The data argue for the existence of either specific terminal DNA sequences for each chromosome tip in *T. pubescens*, or sequences common to all chromosome ends but their extension does not allow detection by in situ hybridization. Both sciarid species share terminal sequences that are significantly underrepresented in chromosome ends of *T. pubescens*. Conclusions: The data suggest an unusual terminal structure in *T. pubescens* chromosomes compared to other dipterans investigated. A putative, evolutionary process of repetitive DNA expansion that acted differentially to shape chromosome ends of the two flies is also discussed.

## 1. Introduction

Telomeric structure composed of short tandem repeats synthesized by telomerase is broadly conserved across taxa. However, telomerase is not present in Diptera [1], an order which displays unexpected diversity in relation to chromosome ends. Telomeric sequences vary even between species of the same genus and provide an example of telomere diversification. Dipteran telomeric structures diverge depending on DNA repeat types and how they are organized at the very end of the chromosomes. In *Drosophila melanogaster*, chromosome termini are composed of specific retrotransposons [2,3,4]. In *D. virilis,* complex terminal satellites, in addition to retrotransposons, were identified at chromosome ends [5,6]. Among lower dipterans (suborder Nematocera), tandem arrays of complex repeats (>120 bp) have been characterized at telomeres of *Chironomus* species [7,8,9,10,11,12], and in *Anopheles gambiae* [13,14].

In the nematoceran *Rhynchosciara americana*, apparently long homopolymer dA/dT tracts [15,16,17,18,19] and reverse transcriptase-related proteins are regularly associated with chromosome ends [18]. These components are uncommon in Diptera and led us to initiate the characterization of terminal sequences in *R. americana* chromosomes. Complex (414 bp) tandem repeats were first characterized in this sciarid fly that were found to be sub-telomeric rather than truly terminal repeats [20]. A second repeat type named M-22 is unusually short for a repeat tandemly arrayed at the chromosome ends of dipterans. Additional data showed that M-22 tandem repeats lie distal to sub-telomeric repeat arrays [21].

Continued investigation of chromosome ends in *R. americana* identified other short tandem repeat types. One composed of 16 nucleotides and named M-16 is frequently intermingled within M-22 arrays. The occurrence of segmental duplications, as deduced by sequence analyses involving islands of M-22 and M-16 repeats that appear to reach the very end of chromosomes, might have implications for the process of chromosome end maintenance in this species [22]. T-14 repeats, only 14 bp long, were cloned eventually in *R. americana*. This is the shortest tandem repeat composing chromosome termini that have been characterized in dipterans to date [23]. M-22, M-16, and T-14 short tandem repeats represent a divergent chromosome end structure compared to long (complex) terminal repeats that have been characterized in chironomids and in *Anopheles*. *D. melanogaster* constitutes a third divergent structure composed of specific mobile elements in place of tandem DNA repeats.

By analyzing distinct *Rhynchosciara* species, the results argue for differential occupation of repetitive sequences at chromosome ends during the evolution of this genus [16,20,21,22,23]. Nevertheless, data restricted to a single genus have not given information on how chromosome ends are structured in other representatives of Sciaridae. Such a limitation also applies to *Chironomus* and *Drosophila*, dipterans that have long been studied with a focus on telomeres. Therefore, insights into the chromosome end evolution within dipteran families through the analysis of distinct genera have been lacking.

Taken the above gap into consideration, *Trichomegalosphys pubescens* was exploited in this work together with *Rhynchosciara americana*, the only sciarid fly that have been studied with a focus on the telomere to date. Relationships within the family Sciaridae have long been controversial. In an early report, *Trichomegalosphys* and *Rhynchosciara* appear in distinct tribes, Megalosphyni and Sciarini respectively [24]. Still considering early classifications, another study placed the two genera in different groups, I and II [25]. More recently, molecular phylogeny as well as cladograms on sciarid relationships have been produced [26] but analyses including *T. pubescens* and *R. americana* have not been available yet. Despite the lack of molecular data, genera placed in distinct tribes or even groups strongly suggest that they are significantly divergent.

Results obtained with homologous as well as heterologous probes indicated that *R. americana* and *T. pubescens*, while having different chromosome end structures, share DNA sequences that appear to be significantly underrepresented in the latter species. The data also suggest two possible structures for chromosome termini in *T. pubescens* as well as a putative process that led to the evolutionary divergence between chromosome ends of *T. pubescens* and *R. americana*.

## 2. Materials and Methods

### 2.1. Animals

Larvae of *R. americana* were collected in the region of Mongaguá, state of São Paulo, Brazil. Larvae were kept in the laboratory at 18–22 °C. *T. pubescens* (Diptera: Sciaridae) larvae were collected either on the campus of the University of São Paulo or in the region of Mogi das Cruzes, state of São Paulo Brazil. This species was formerly named *Trichosia pubescens* [27]. Laboratory cultures were kept at 18–22 °C to a 12/12 h light/dark cycle.

### 2.2. Preparation of Chromosome Spreads

Salivary glands were dissected in Ringer buffer or 1×PBS and briefly fixed in ethanol-acetic acid (3:1). After squashing the salivary glands in 50% acetic acid, the slides were frozen on dry ice for 10 min. The coverslips were pried off with a razor blade and the slides were then kept in absolute ethanol at −20 °C until microdissection or hybridization procedures. Chromosome spreads of *Chironomus riparius* (Diptera: Chironomidae) were prepared as a control as described above and were kindly sent by Dr. J.L. Martinez-Guitarte. This species was formerly named *Chironomus thummi thummi* [28].

### 2.3. Microdissection

Chromosome ends were microdissected with the aid of microneedles made from solid glass tubes with the aid of a microforge (De Fonbrune). Microdissection was carried out on a Zeiss Axiovert S100 inverted microscope coupled with a manual micro-manipulator (Narishige). The microdissected fragment was recovered from air-dried slides with the tip of a micro-needle that was subsequently broken into a polymerase chain reaction (PCR) tube.

### 2.4. DOP-PCR

The DNA in the chromosome fragment was amplified by polymerase chain reaction (PCR) using degenerate oligonucleotides (DOP-PCR) [29] in a thermocycler (*Mastercycler*, *Eppendorf*), omitting topoisomerase and proteinase K treatments. The following reaction mixture was added to the PCR tube: 0.2 mM of each dNTP; 1× Taq buffer; 2.5 U Taq DNA polymerase (Life Tech, Carlsbad, CA, USA); 1 mM MgCl_2_; 75 pM DOP primer, H_2_O to a final volume of 50 µL. Prior to the amplification rounds, the mixture was briefly centrifuged. Cycle conditions for the first amplification were 93 °C for 4 min followed by 8 cycles of 94 °C for 1 min, 30 °C for 90 s, and 72 °C for 3 min. The first amplification was completed with additional 28 cycles of 94 °C for 1 min, 56 °C for 1 min, 72 °C for 3 min. For the second amplification round, 5 µL from the volume of the first amplification reaction were added to 50 µL of a new reaction mixture exactly as described above. Cycle conditions were 93 °C for 4 min followed by 26 cycles of 94 °C for 1 min, 56 °C for 1 min, 72 °C for 3 min, and a final extension at 72 °C for 5 min.

The amplification products were checked in agarose gels before cloning or labelling procedures.

### 2.5. Cloning and Sequencing of PCR Products

The purified PCR products were ligated in pGEM-T Easy vector according to standard protocols (Promega, Madison, WI, USA). Procedures for transformation and plating of *E. coli* DH5 strain were described elsewhere [30]. Individual colonies (134) were randomly chosen to grow in liquid medium for plasmid DNA extraction using Concert Rapid Plasmid Miniprep System (Life Tech) to be subsequently analyzed by Southern-blot hybridization using salivary gland DNA of *T. pubescens* as a probe. Sequencing reactions of both strands of plasmid DNAs were performed with Big Dye Terminator Cycle Sequencing Ready Reaction as recommended by the manufacturer (Applied Biosystems, Foster City, CA, USA) using M13 Forward or Reverse primers (Life Tech) and subsequently run in the ABI PRISM 310 Genetic Analyzer (Applied Biosystems). GenBank searches were done with BLAST [31].

### 2.6. Southern-Blot Hybridization

Plasmid or genomic DNA were then cut with *Eco*RI, the digests run in agarose gels prepared according to current protocols [30] and transferred to Hybond N+ or Zeta Probe membranes according to standard procedures (Bio-Rad). Salivary gland DNA (200 ng) and p*Tp*-36 insert (50 ng) were labelled by random priming with ^32^P-dATP following current protocols (Life Tech). Hybridization was carried out overnight at 60 °C in 0.5 M Na_2_HPO_4_, 2% SDS. The membranes were washed twice at 60 °C for 30 min in 40 mM Na_2_HPO_4_, 2% SDS. Restriction enzymes and DNA size markers were purchased from New England Biolabs.

### 2.7. Non-Radioactive Labelling of the Probes

Probe labelling was done with 0.1 mM biotinylated-11 dUTP (Sigma, Burlington, MA, USA) replacing dTTP in 50 µL of PCR reaction mixture as described previously for the second amplification round of DOP-PCR using 2 µL of total DOP-PCR products. P*Tp*-4 plasmid insert was labelled replacing DOPs by M-13 Forward and Reverse primers (Life Tech). A second probe derived from the p*Tp*-4 plasmid was made as described above with one modification. Forward and reverse primers were replaced by two primers (Tp-4a and Tp-4b) flanking the tandem array contained in the insert. Tp-4a: 5′GTTGGAAATAATCTTTAACC3′. Tp4-4b: 5′AATAGTGTTCACTTTCCTTTTC3′. Plasmids (p*Tp*-1, p*Tp*-2, p*Tp*-3) and phage DNA were labelled by nick translation using DIG Nick Translation Mix (Roche, Basel, Switzerland) according to the instructions of the manufacturer.

### 2.8. In Situ Hybridization

The probe mixture consisted of labelling reaction product (50 µL), 40% formamide, 2X SSC and 0.1% SDS to a final volume of 110 µL. The probe mixture (5–10 µL) was applied on each air-dried slide and covered with a plastic coverslip. The slides were steam heated at 75 °C for 5–10 min and immediately kept in a closed box. Hybridization was carried out overnight at either 32 °C or 37 °C. The slides were then washed in 0.5× SSC, 0.2% SDS at either 32 °C or 37 °C for 30 min, followed by incubation at room temperature in 1× TBS, 0.1% Triton X-100 (TBST) for 10–20 min. For fluorescent detection, goat IgG anti-biotin (Sigma) diluted 1:50 in TBST was applied on the slides for 1 h at room temperature followed by washes in TBST and incubation for 45 min with rhodamine-labelled rabbit IgG anti-goat (Sigma) diluted 1:100 in TBST. Chromosomal DNA was stained with 4′,6-diamidino-2-phenylindole (DAPI). For detection with alkalyne phosphatase, slides were incubated with anti-digoxigenin conjugated with alkalyne phosphatase (Roche) diluted 1:100 in TBST for 1 h at room temperature. The slides were then washed in TBST for 10 min and finally in 1× TBS for 5 min. For fluorescence inspection, the slides were mounted in antifading medium (Vectashield, Vector Labs, Burlingame, CA, USA) and inspected under a Nikon Eclipse 80i microscope equipped with epifluorescence optics. Images were captured using the NIS-Elements software package (Nikon, Tokyo, Japan).

### 2.9. Genomic Library

The *T. pubescens* genomic library, kindly provided by Dr. E.M.B. Dessen, was prepared with DNA from *T. pubescens* salivary glands using Lambda EMBL4 as a vector. Procedures for screening, phage DNA extraction, and sub-cloning of restriction fragments were carried out according to usual protocols [30].

## 3. Results

### 3.1. Polytene Chromosomes of R. americana and T. pubescens

Chromosome description of the species studied in this work is limited in the literature, particularly for *T. pubescens* [32]. For this reason, the reintroduction of the polytene chromosomes of the two sciarid species enables comparisons of how evolutionary change shaped the chromosomes of *R. americana* and *T. pubescens*. Three of the four polytene chromosomes of *R. americana* (B, C, and X) display proximal heterochromatin blocks within sections B-15, C-11, and X-12 that hybridize to rDNA probes [33]. Polytene chromosomes of this species have thus three centromeric or pericentric heterochromatic ends and five non-centromeric ends, corresponding to the sections A-1, A-18, B-1, C-1, and X-1 (Figure 1).

Although *T. pubescens* also has four polytene chromosomes, there are significant structural differences in relation to *Rhynchosciara*. *T. pubescens* chromosomes lack centromeric or pericentric ends as well as conspicuous heterochromatic blocks associated with these regions as seen in *Rhynchosciara* species. Polytene pericentric regions of *T. pubescens* are usually identified by chromosome breakpoints as a result of local DNA under-replication (Figure 2).

In *T. pubescens*, ribosomal DNA (rDNA) probes hybridized at the mid portion of the chromosome X, very close to the breakpoint of the polytene X chromosome [33]. Hence, centromeres and pericentric regions occupy different positions in chromosomes of *Rhynchosciara* and *T. pubecens* in addition to being morphologically distinct.

### 3.2. Localization of DNA Sequences from the B-1 Tip of R. americana in R. americana Chromosomes

The results of in situ hybridization were obtained with a probe synthesized with DNA from a single B-1 polytene chromosome tip of this species and are presented for the first time. Significant hybridization signals were seen in all telomeres but displaying variable signal intensity at certain chromosome ends. A-18, B-1, and C-1 tips always appeared more intensely labelled than A-1 and X-1 tips. Hybridization was also detected at centromeric ends, pericentric regions of all chromosomes and at several interstitial sites (Figure 3).

Use of individual microdissected polytene chromosome ends other than B-1 has resulted in probes that are able to reproduce essentially the same results as observed in Figure 3, namely the probe from any single chromosome end hybridized to the five non centromeric ends of this species.

### 3.3. Localization of DNA Sequences from Chromosome Tips of T. pubescens in T. pubescens Chromosomes

The experiments performed in this species were initiated with a probe synthesized by PCR from a single X-1 tip of *T. pubescens* that was labelled and used for in situ hybridization. In contrast to the results obtained in *R. americana* chromosomes with the B-1 probe described above, no hybridization signal was seen at chromosome ends other than X-1. Labelling of lower intensity was also observed in chromosome breakpoints and at interstitial sites (Figure 4).

Given the results in disagreement with what is usually expected, microdissection procedures were repeated in order to discard technical artifacts. The results obtained with probes from distinct chromosome ends reproduced the same hybridization pattern as shown in Figure 4, namely there is a main signal corresponding to the microdissected end but no significant labelling, if any, was seen at other chromosome termini. The possible DNA requirement from more than one microdissected chromosome end was considered as all the *T. pubescens* probes, as done in *R. americana*, were made with a single chromosome tip. Use of a probe made of two distinct tips of *T. pubescens* chromosomes resulted in labelling restricted to two chromosome ends corresponding to microdissected tips used for the probe synthesis (Figure 5).

### 3.4. Localization of Heterologous Probes in T. pubescens and R. americana Chromosomes

Although *T. pubescens* and *R. americana* are distantly related sciarid flies, and in addition probes from microdissection produced disparate hybridization results, the possibility that probes from chromosome tips of *T. pubescens* could hybridize to *R. americana* chromosomes was considered. The probe made with A-1 tip DNA of *T. pubescens* was clearly localized at all non-centromeric ends of *R. americana*. Significant labelling was also observed in the pericentric heterochromatin and interstitial bands in all the experiments performed (Figure 6).

Results obtained with the probe from the X-1 end of *T. pubescens* on chromosomes of *R. americana* were essentially those obtained with the A-1 tip probe of *T. pubescens* although producing less intense signals and also distinct localizations with regard to interstitial chromosome sites (Appendix A).

The results described above showed that *T. pubescens* and *R. americana* share sequences at chromosome ends as well as in other genomic regions despite divergent labelling patterns in chromosomes of the two species. For this reason, the possibility that the *R. americana* B-1 probe could hybridize to chromosomes of *T. pubescens* was tested. In the first attempts, weak labelling was observed in polytene chromosomes of *T. pubescens* when hybridization and washes were performed at 37 °C. However, when the hybridization temperature and washings was lowered to 32 °C, high intensity signals were detected at B-25 ends. Less intense staining was also observed at interstitial regions of the chromosomes of this species (Figure 7).

The above results raised the possibility that repeats from *R. americana* chromosome ends characterized previously might be detected in *T. pubescens* chromosomes under low stringent hybridization and washings. Nevertheless, use of the *R. americana* sub-telomeric DNA probe [20] and terminal repeats localized distally [21,22,23] did not produce results in *T. pubescens* chromosomes even by lowering the temperature of hybridization and washings. Controls using *R. americana* polytene chromosomes showed that the probes used for hybridization were labelled and reproduced in this species the expected results as already published.

### 3.5. Controls for Microdissection Procedures

Microdissection procedures involving chromosomes of *R. americana* and *T. pubescens* were separated by a five-year interval in order to discard the possibility of contamination and always performed with newly sterilized material. When the results led to suspect that either contamination or another methodological problem had occurred, additional microdissections were carried out using a new reagent set. The results of in situ hybridization presented in this work were reproduced in a large number of slides.

Another control was introduced in this work by performing in situ hybridization using a probe obtained by DOP-PCR in chromosomes of *Chironomus riparius* (Diptera: Chironomidae), a nematoceran that has extensively been studied with a focus on the telomere. Telomeric puffing induced by heat shock is a very rare phenomenon that has been observed in polytene chromosomes of *C. riparius* [34]. The efficacy of the methods employed in this work was assessed by microdissecting a single terminal puff (TBRIII) of this species for use in DOP-PCR and subsequent probe preparation. The ability of the above procedure, employed for the first time in this species, to amplify telomeric sequences was confirmed since the probe synthesized from a single TBRIII DNA hybridized to the expected seven non-centromeric ends of *C. riparius*. Interstitial bands and certain pericentromeric regions appeared labelled less intensely than telomeres, probably indicating that non-telomeric DNA sequences were microdissected together with TBRIII DNA (Figure 8).

### 3.6. Screening of a Plasmid Library Constructed with X-1 Tip DNA of T. pubescens

DOP-PCR performed with DNA from the X-1 tip of *T. pubescens* was used for the construction of a plasmid library. Plasmid DNA from 134 colonies was digested with *Eco*RI and analyzed by Southern-blot hybridization using salivary gland DNA of *T. pubescens* as the probe. Controls included ribosomal DNA (rDNA) inserts as references for moderately repetitive sequences. Such a procedure was employed in previous reports [7,8,20,21,22,23,35] and its rationale exploits the genomic representation of repetitive DNA. Total genomic DNA labelled for hybridization is enriched with highly repetitive DNA. Therefore, it would be able to produce, by comparison, to single copy or even middle repetitive DNA, significant hybridization signals in plasmid inserts representing highly repetitive sequences.

No hybridization signal indicating the existence of cloned inserts with highly repetitive sequences was detected in *T. pubescens*, only two inserts showed signal intensity comparable to middle repetitive DNA. In addition, in situ hybridization results using these short inserts showed no significant signal in polytene chromosomes of either *T. pubescens* or in *R. americana*, reinforcing that they do not correspond to repetitive DNA in the genomes of the two species. Inserts that displayed hybridization signals apparently representing middle repetitive DNA together with others that were randomly chosen were sequenced. Most results showed no significant similarity with sequences deposited in data banks except for the two inserts containing features described above (Appendix A). The p*Tp*-29 insert is 67% similar to a stretch identified in the genomic chimera named “RaTART” analyzed in detail previously [35], which in turn displays high identity with sequences encoding reverse transcriptase of insect mobile elements.

The second insert, p*Tp*-36, is 97% similar to that found in the p*Ra*-43 insert from the chromosome end of *R. americana* [21]. The two sequences contain short microssatellite arrays and also a 47 bp repeat unit named M-47 that is not tandemly organized in the *R. americana* genome. It was characterized as sub-telomeric in a previous work and appeared enriched in chromosome ends A-18 and X-1 of *R. americana* [21]. The size and sequences of p*Tp*-36 and p*Ra*-43 inserts are not exactly identical, and the two libraries were constructed with a time separation of five years. The micro-libraries of the two species have also never been manipulated simultaneously and the p*Tp*-36 insert hybridized to genomic DNA of *T. pubescens* during the screening of the microlibrary. An additional control was made by performing Southern-blot hybridization under high stringent conditions using salivary gland DNA of *T. pubescens* and the p*Tp*-36 as a probe (data not shown). These results confirmed that sequences within the p*Tp*-36 insert are present in the *T. pubescens* genome.

### 3.7. Screening of a Genomic Library of T. pubescens with pTp-36 Insert

As the p*Tp*-36 insert was the only cloned DNA that represented a DNA stretch shared by chromosome ends of *T. pubescens* and *R. americana*, it was used as a probe to isolate genomic clones from a phage library in an attempt to provide information on the structure of telomeres/sub-telomeres of *T. pubescens*. Four genomic clones were isolated and produced multiple hybridization signals in chromosomes of *T. pubescens* (data not shown). Attention was given to the phage 2 insert that was the only genomic clone that also produced hybridization signals in chromosomes of *R. americana* (Appendix A). Phage 2 DNA was cut with restriction enzymes and four fragments were sub-cloned for further analyses.

After sequencing two sub-clones, p*Tp2.*2 and p*Tp2.*3, no significant similarity with sequences deposited in databases was observed. In addition, both sub-clones produced very weak hybridization signals dispersed in all chromosomes of *T. pubescens*, showing no preferential localization at chromosome ends that were devoid of hybridization signals. Both probes did not produce hybridization signals in chromosomes of *R. americana*. Therefore, as the two sub-clones did not contribute with information to the aim of this investigation, they were not included as figures.

Hybridization results using the p*Tp*-2.1 were also included in this report since part of its sequence could be identified in databases and, in addition, very clear hybridization results were obtained in polytene chromosomes of *T. pubescens*, displaying localization in multiple chromosome sites, although no enrichment at chromosome ends was observed (Appendix A). Conceptual translation of part of its sequence using BLASTX showed identity with structural proteins from densoviruses (Appendix A).

At last, after sequencing the whole p*Tp*-2.4 insert, no similarity with sequences deposited in data banks was found. However, a visual inspection showed a stretch that seemed to contain repetitive DNA (Figure 9a). A manual alignment of sequences composing repeat candidates was made and enabled the identification of a 31 bp repeat unit organized as a tandem array (Figure 9b), named M-31.

The p*Tp*-2.4 plasmid was labelled and then used for in situ hybridization in polytene chromosomes of *T. pubescens*. The most intense hybridization signals were detected in several interstitial chromosome regions namely, no apparent preference for chromosome ends was observed (Figure 10).

As the M-31 represents a novel satellite-like sciarid tandem repeat, the probe was also used in chromosomes of *R. americana* to check its possible occurrence in the genome of the latter species. Interestingly, the probe hybridized to the *R. americana* chromosomes showing the most intense signals restricted to chromosome ends A-18 and X-1 of this species and also weak labelling dispersed by interstitial sites (Figure 11).

In order to discard that the probe signals came from sequences that are not contained in the tandem array of the p*Tp*-2.4 insert, a second probe was made by PCR using the same plasmid, but instead two primers flanking the tandem repeat so that other sequences were excluded from the probe. The hybridization results obtained in polytene chromosomes of the two species were the same as those showed in Figure 10 and Figure 11, confirming that chromosomal location of the probe corresponds specifically to the tandem repeat in the p*Tp*-2.4.

## 4. Discussion

Given the diversity of sequences composing chromosome termini in Diptera, massive sequencing methods cannot help to provide information on chromosome ends of species in this order. Microdissection followed by cloning procedures have been proven to be useful to this aim and have long been employed in telomere biology of nematocerans [8]. This report represents the first attempt to compare chromosome ends of two genera that belong to the same dipteran family using the procedures described above.

Probes obtained by microdissection of a single chromosome tip of *R. americana* indicated that sequences are shared by non-centromeric ends of this species. Localization results that were seen in the past, but showed for the first time in this work, allowed the characterization of repeats composing telomeres and sub-telomeres [20,21,22,23] from micro-libraries constructed with microdissected DNA. Tandem DNA repeats were found to be shared by all non-centromeric ends of *R. americana.* In this sense, *R. americana* is not an exception since terminal repeats of chironomids were also found to be specific to non-centromeric ends [7,8,9,10,11,12].

Nevertheless, when the same methodological approach was applied to *T. pubescens*, the results unexpectedly showed that sequences within a given chromosome tip do not seem to be present at other chromosome ends. The data allow to suggest two possibilities for the structure of chromosome termini of this species. The first possibility could be seen as quite unusual and implies specific sequences for each chromosome tip. The second, more conservative possibility argues for the existence of DNA sequences common to all chromosome ends in *T. pubescens.* However*,* the short extension of these sequences impedes detection in chromosome ends by in situ hybridization. In any case, the data imply an unusual chromosome end structure *T. pubescens* in relation to dipteran data available in the literature.

Positive results of heterologous probe hybridization, although showing very different localization patterns, argue for sequence conservation to variable extents within chromosome ends of *R. americana* and *T. pubescens*. Since consistent labelling by the *T. pubescens* probe occurred in non-centromeric ends, pericentric regions and interstitial sites of *R. americana,* the data strongly suggest in addition that *T. pubescens* sequences from a single chromosome tip are over-represented in the genome of *R. americana*. On the other hand, the *R. americana* B-1 tip probe produced hybridization results that were preferentially detected at B-25 ends of *T. pubescens* by lowering the hybridization stringency. These data argue that probe and genomic sequences are dissimilar and, in addition, that B-1 and B-25 ends are syntenic. Whether the putative short extension of telomeric DNA in *T. pubescens* does not allow detection by in situ hybridization as hypothesized above, the synteny suggested would be representative of sub-telomeric DNA that also seems specific for each chromosome end of *T. pubescens* and conserved to some extent at B-1 tip of *R. americana*. As homologous and heterologous probes used separately are predominantly located respectively in chromosomes X and B of *T. pubescens*, these features represent further support for the peculiar structure of chromosome ends in this species.

A comparison with data from the screening of the *R. americana* library made in the past with a single B-1 tip DNA may give clues about differences in the chromosome end structure of the two sciarid species. Three inserts displaying a high degree of reiteration were found within 54 inserts from the B-1 micro-library of *R. americana* that were chosen at random [20]. Another insert from the same micro-library (p*Ra*-43) representing moderately repetitive DNA was characterized eventually [21]. On the other hand, no highly repetitive DNA insert was found after screening more than 100 inserts from the *T. pubescens* microlibrary that were also chosen by chance, only two representing middle repetitive DNA were identified. The data from the two screenings suggest that the representation of repetitive DNA at chromosome ends of *T. pubescens* is significantly lower than that of terminal regions of *R. americana* chromosomes. Such an assumption agrees with in situ hybridization results using *T. pubescens* probes obtained by microdissection and reinforces the divergent telomere/sub-telomere structure in *T. pubescens* compared to those that have been characterized in flies up to now.

Among the results obtained after screening the genomic library of *T. pubescens*, the identification of a plasmid insert (p*Tp*-2.1) containing sequences related to densovirus proteins is possibly far from helping in the characterization of sequences composing chromosome ends of *T. pubescens*. Densoviruses have been described in a number of organisms but insertion of viral DNA in genomes has been seldom reported [36]. In situ hybridization data in *T. pubescens* using p*Tp*-2.1 as probe were included in this report because they are likely to be the first chromosomal visualization of densovirus DNA insertion in a dipteran genome. The second plasmid of interest from the screening (p*Tp*-2.4) contains a tandem repeat that is a novel minisatelite-like sequence in sciarid genomes. Named M-31, it is the first repeat characterized in *T. pubescens* and, in addition, the first tandem repeat detected by in situ hybridization in two sciarid genera. M-31 arrays appeared in many chromosome regions of *T. pubescens* without displaying preferential localization in the genome of this species. In contrast, M-31 repeats are clearly enriched in chromosome ends A-18 and X-1 of *R. americana* that also correspond to the main location of M-47, sub-telomeric repeats described previously in this species [21]. Interestingly, the M-47 sequence is part of the p*Tp*-36 insert used to screen the genomic library of *T. pubescens* and made possible the characterization of M-31 repeats.

M-31 arrays were detected in certain chromosome ends of *T. pubescens*. However, if the hypothesis on short tandem repeats in genomes that are below the detection limits is considered, the occurrence of M-31 arrays at chromosome ends devoid of labelling remains conjectural. Despite this apparent lack of M-31 DNA in some chromosome ends of *T. pubescens*, the disparate chromosomal localization of M-31 in the two species studied in this work together with the data that have been obtained in this work suggest that *T. pubescens* and *R. americana* genomes underwent an evolutionary process that led to divergent chromosome end structures in the two species as hypothesized in the following paragraphs.

Sequences from a single chromosome end of *T. pubescens* that were found to be over-represented in *R. americana* indicate a repetitive DNA expansion in the latter species. This process seems to have acted more particularly at chromosome ends and in proximal heterochromatin of *R. americana* according to the in situ hybridization data. In *T. pubescens,* localization results using homologous and heterologous chromosome tip probes suggest that repeat expansion has been inactive or may have acted with extremely low intensity during the evolution of this species.

Chromosome end repeats of *R. americana*, characterized prior to this work, have not been detected by in situ hybridization in *T. pubescens* chromosomes. The results argue for the absence of these sequences in the *T. pubescens* genome or, alternatively, for their genomic under-representation, thus impeding visualization by hybridization in chromosomes of this species. Sequencing results from the screening of *R. americana* and *T. pubescens* micro-libraries constructed with chromosome end DNA also point to a higher representation of repetitive DNA at chromosome ends of *R. americana*. These data reinforce preferential repeat expansion at chromosome ends of *R. americana*.

The p*Tp*-36 insert, from the chromosome end X-1 of *T. pubescens*, could not be detected by in situ hybridization in its expected genomic location. In contrast, the p*Ra*-43 insert, displaying highly similar sequences and characterized previously in *R. americana*, was detected at chromosome ends A-18 and X-1 of this species [21]. It is worth mentioning that the p*Ra*-43 insert, whose origin is the microdissected chromosome tip B-1 of *R. americana*, did not produce hybridization signals at chromosome end B-1 of this species. The data argue for repeat expansion as a requirement for detection by in situ hybridization in the two species studied in this work and moreover converge to the repeat expansion hypothesis in chromosome ends of *R. americana*.

M-31 repeats characterized in this work display preferential localization in interstitial sites of *T. pubescens* chromosomes as inferred by in situ hybridization results, not being detected at all chromosome ends of this species. In contrast, M-31 arrays appeared clearly concentrated at chromosome ends A-18 and X-1 of *R. americana*. The data represent further support for an evolutionary process of repeat expansion, toward accumulating tandem and dispersed DNA repeats at chromosome ends of *R. americana*.

## 5. Conclusions

In summary, there are sequences shared by chromosome ends of two distantly related sciarid genera. Part of these sequences may be dissimilar and are differentially represented in both genomes, particularly at chromosome ends of the two species by the action of a putative process expanding locally repetitive DNA in the genome of *R. americana*. Compared to other organisms displaying non-canonical chromosome ends, the results introduce an unusual terminal structure in *T. pubescens* chromosomes as synthesized by two hypotheses. One argues for terminal DNA sequences that are specific for each chromosome tip, being absent in other chromosome termini. According to the second hypothesis, there are DNA sequences common to all chromosome ends in *T. pubescens* but DNA extension limits prevent detection by in situ hybridization. In any case, how the two putative structures work to counterbalance the terminal DNA loss in the absence of telomerase, remains to be elucidated.

By exploiting a single sciarid species, *T. pubescens*, this investigation indicates that chromosome end diversification may occur in distinct genera within a dipteran family. Flies that have been studied with a focus on chromosome ends are far from representing the organism diversity within this order. To broaden this scenario, basic studies are necessary in order to keep other dipterans in the laboratory, a task that is not always possible. An additional issue that persists is that cytogenetic methods necessary for telomere research cannot always be performed in several species. However, there are genera still unexploited within Drosophilidae, Chironomidae, Culicidae, Psychodidae, and Sciaridae that can be raised in the laboratory and also allow chromosome research. The introduction of new dipteran genera in chromosome end studies may show structures other than those that have already been characterized to date.

## Figures and Tables

**Figure 1 cells-09-02425-f001:**
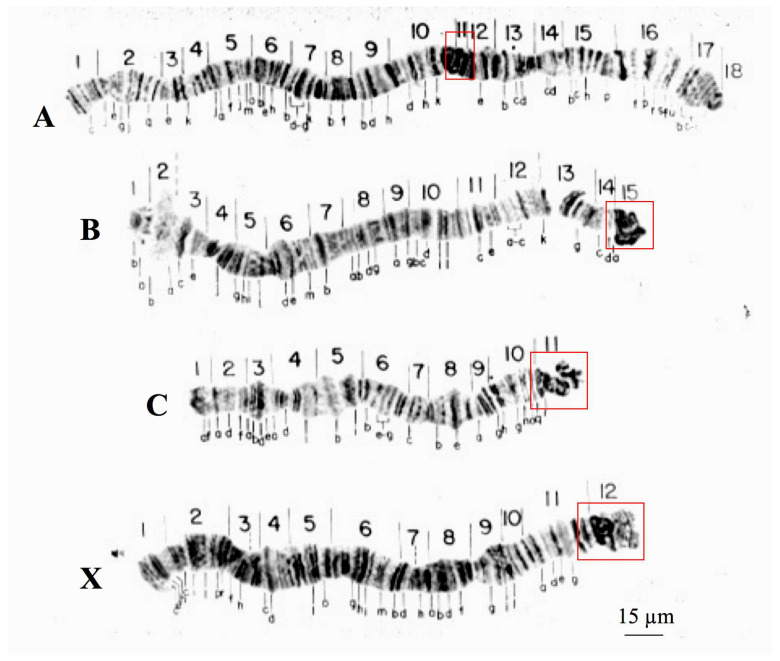
Polytene chromosomes of *R. americana* modified from [16]. Boxes (red) are signaling chromosome sections comprising centromeric and pericentric regions. Chromosome nomenclature (**A**–**D**) as well as section numbering was described previously [16].

**Figure 2 cells-09-02425-f002:**
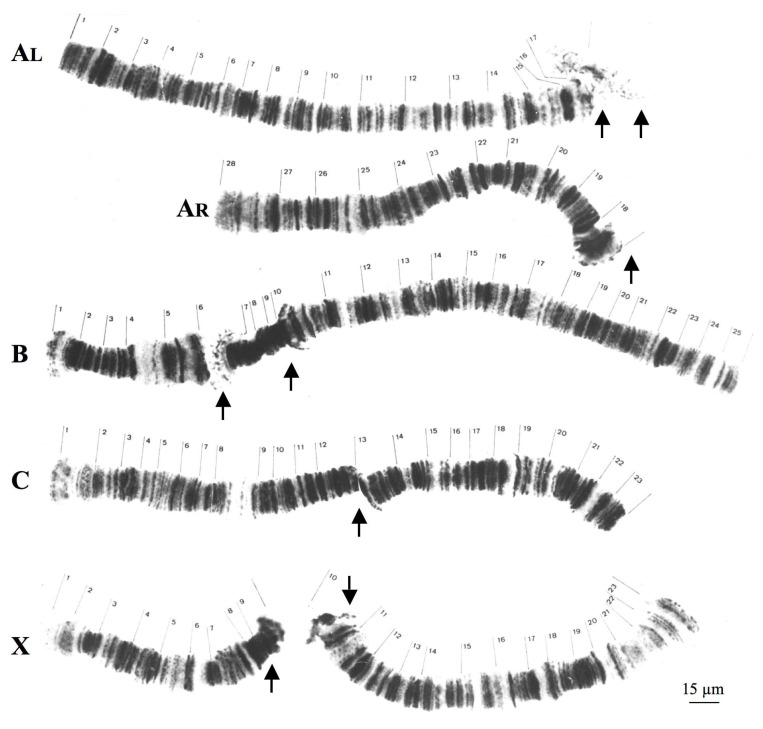
Polytene chromosomes of *T. pubescens* modified from [32]. *AL* and *AR* correspond respectively to long and short arms of chromosome A. The arrows indicate the most frequent breakpoint regions in each chromosome. Chromosome nomenclature (**A**–**D**) as well as section numbering was described previously [32].

**Figure 3 cells-09-02425-f003:**
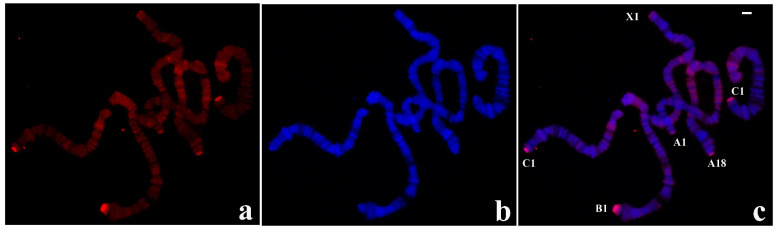
(**a**) Localization of the probe (red signal) synthesized with DNA from a microdissected B-1 chromosome end of *R. americana* in polytene chromosomes of *R. americana.* (**b**) The corresponding image stained with DAPI (blue signal) and (**c**) the merged signals. Non centromeric chromosome ends labelled by the probe were identified. Bar = 20 µm.

**Figure 4 cells-09-02425-f004:**
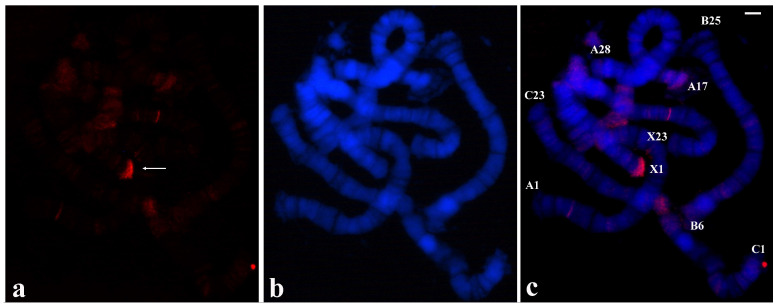
(**a**) Polytene chromosomes of *T. pubescens* after in situ hybridization (red signal) using the probe made of DNA from the microdissected X-1 chromosome end of *T. pubescens.* The arrow points to the main hybridization signal at the chromosome end X-1. (**b**) The same chromosomes stained with DAPI and (**c**) the merged signals. Bar =15 µm.

**Figure 5 cells-09-02425-f005:**
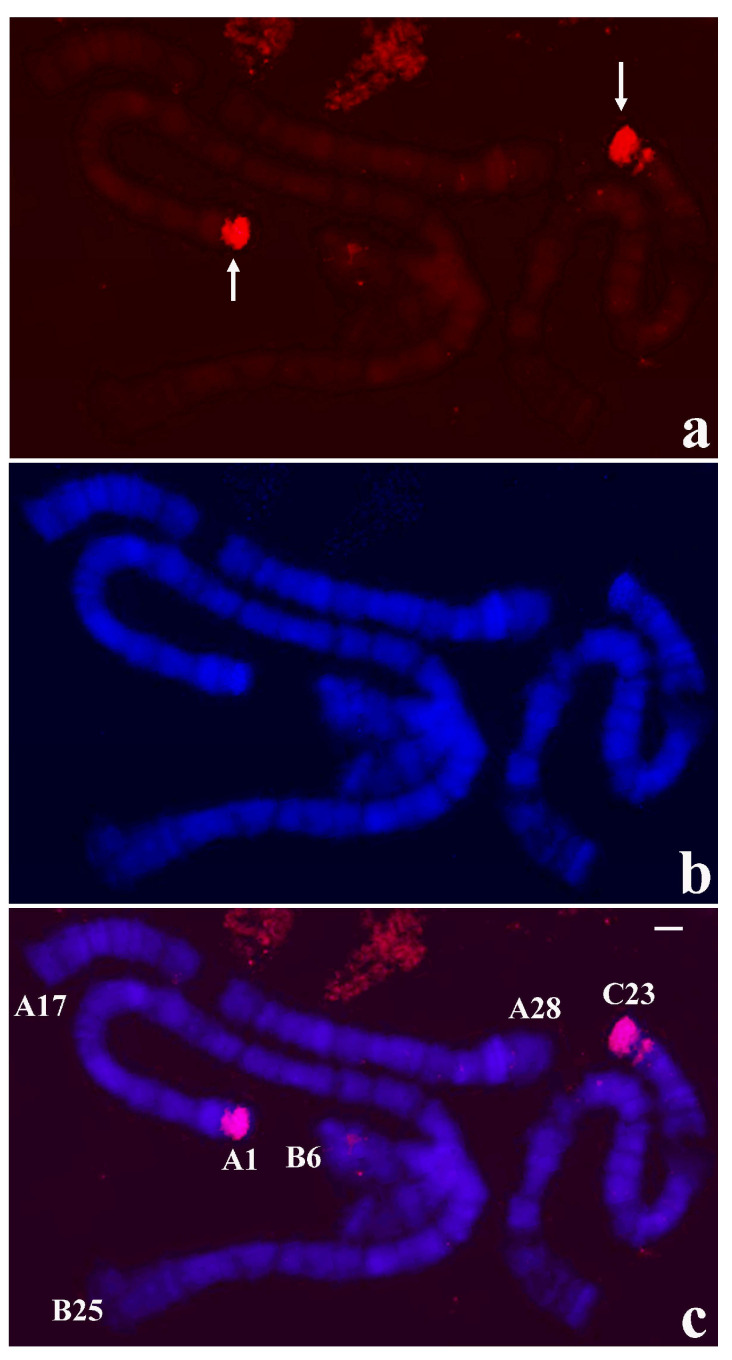
(**a**) Localization of the probe (red signal) synthesized with DNA from two microdissected chromosome ends A-1 and C-23 of *T. pubescens* in polytene chromosomes of *T. pubescens*. The arrows point to the main hybridization signals. (**b**) Chromosomes stained with DAPI and (**c**) the corresponding merged signals. Image superimposition shows chromosome end labelling at A-1 and C-23. Bar = 15 µm.

**Figure 6 cells-09-02425-f006:**
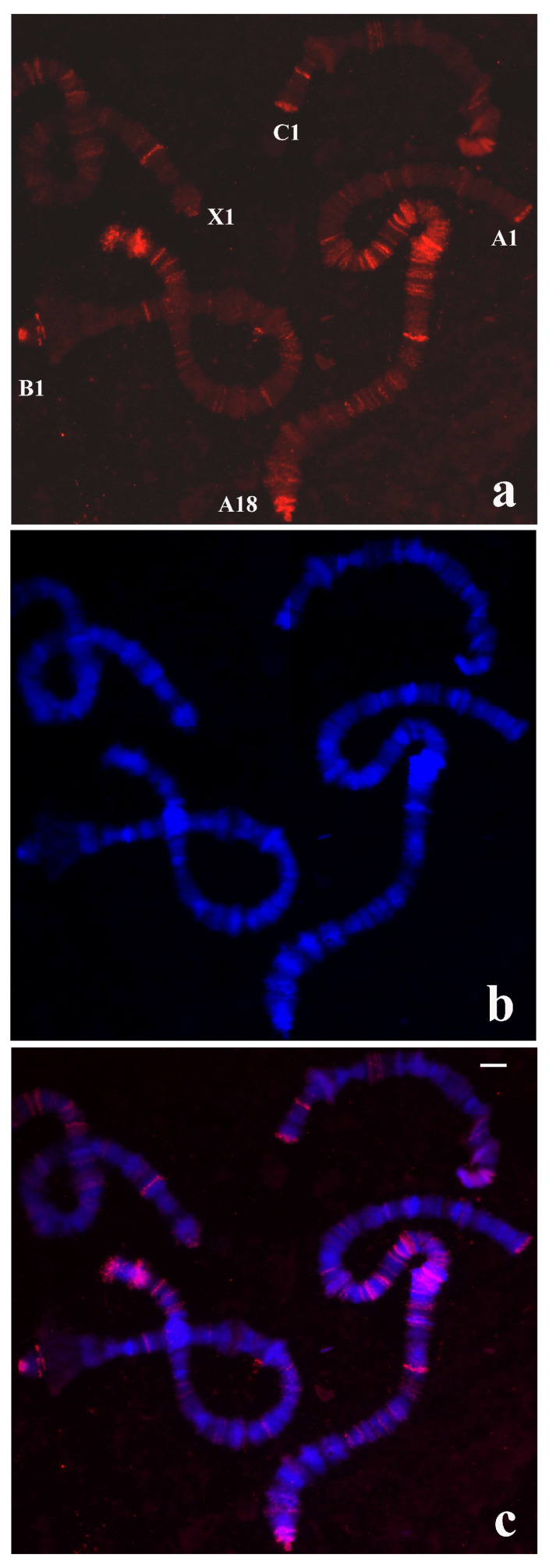
(**a**) Localization of the probe (red signal) synthesized with DNA from a microdissected chromosome end A-1 of *T. pubescens* in polytene chromosomes of *R. americana*. (**b**) Chromosomes stained with DAPI and (**c**) the corresponding merged signals. Chromosome ends labelled by the probe are identified. Bar = 20 µm.

**Figure 7 cells-09-02425-f007:**
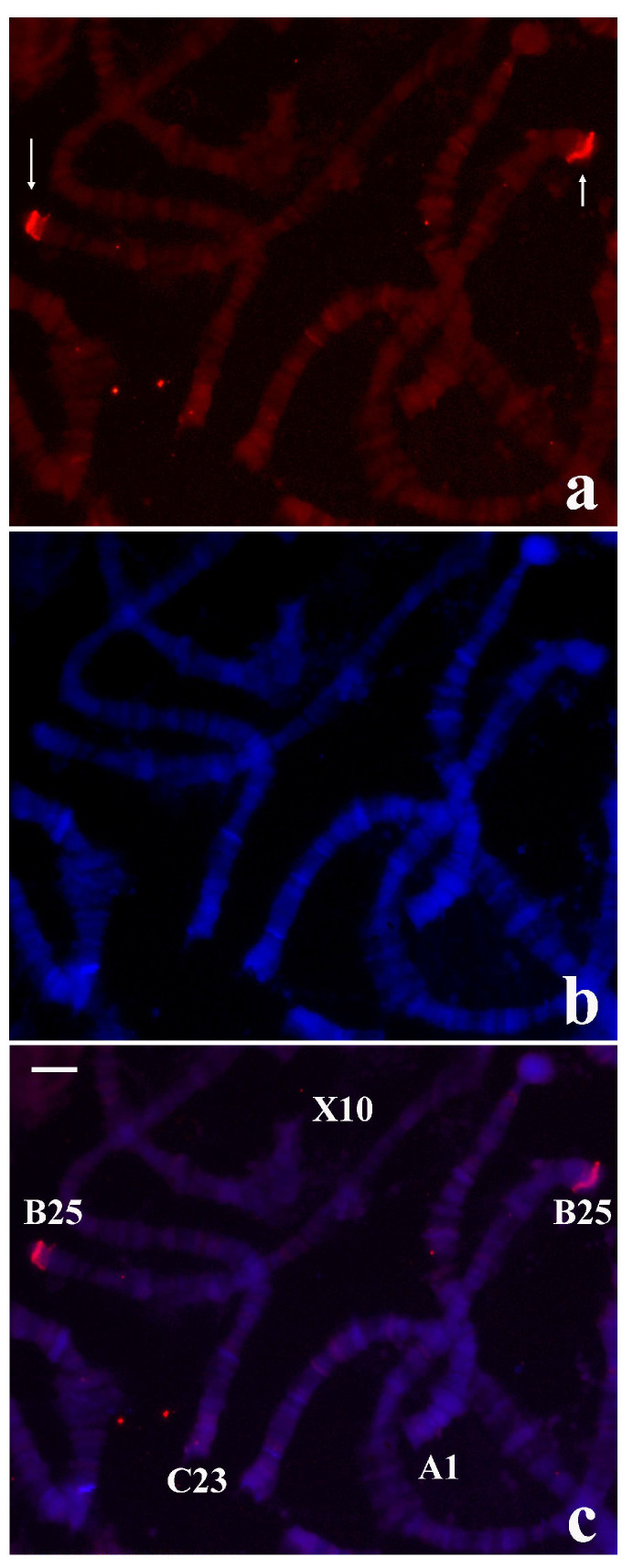
(**a**) Localization of the probe (red signal) synthesized with DNA from the microdissected B-1 tip of *R. americana* in chromosomes of *T. pubescens.* (**b**) The same chromosomes stained with DAPI and (**c**) the corresponding merged signals. The arrows point to the main labelling restricted to chromosome ends B-25. Bar = 15 µm.

**Figure 8 cells-09-02425-f008:**
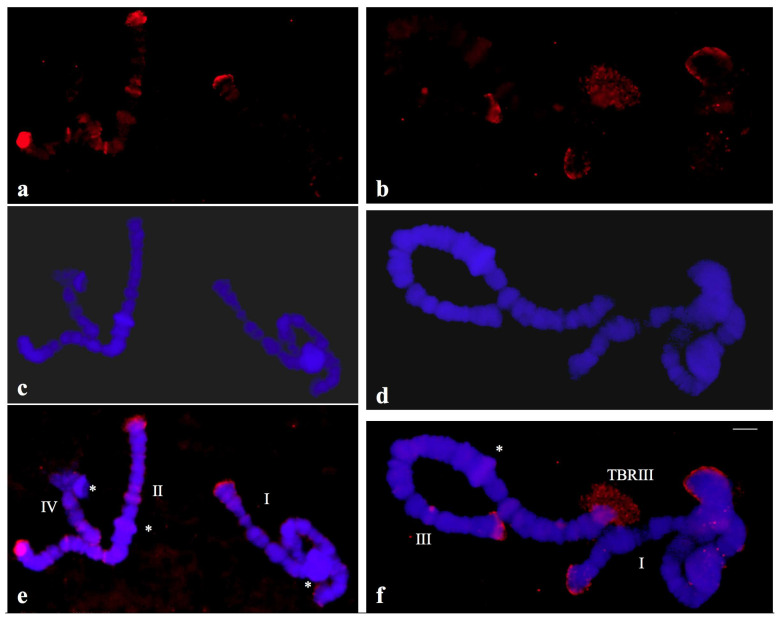
(**a**,**b**) Localization of the probe (red signal) synthesized with DNA from a microdissected telomeric puff (TBRIII) of *Chironomus riparius* in polytene chromosomes of *C. riparius.* (**c**,**d**) the same chromosomes stained with DAPI and (**e**,**f**) the merged hybridization and DAPI signals. The asterisks indicate the centromeric regions of the chromosomes identified individually (I, II, III, IV).

**Figure 9 cells-09-02425-f009:**
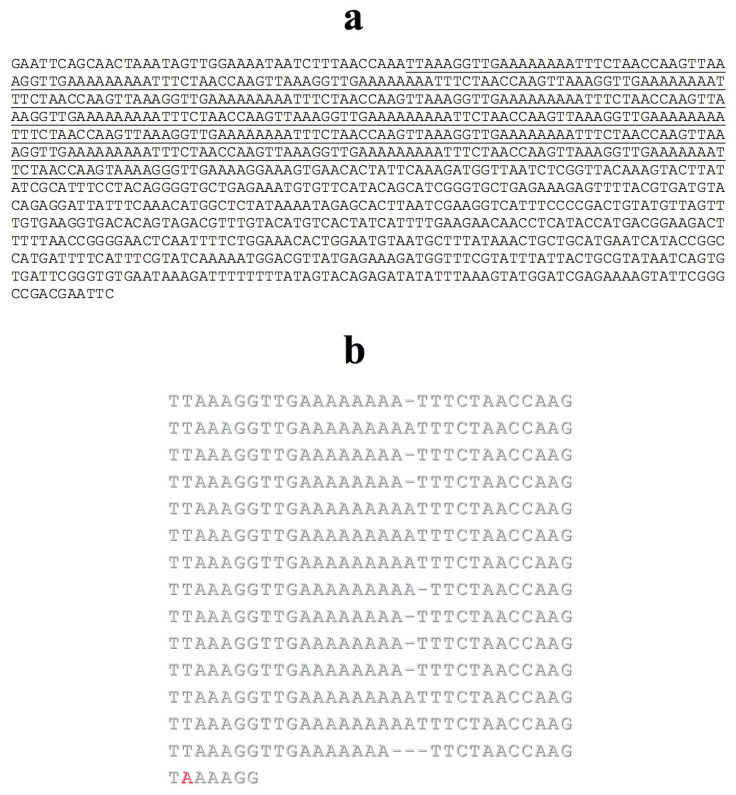
p*Tp*2.4 insert sequence (**a**); stretches comprising tandem repeats are underlined. (**b**) The manual alignment was done with repeats taken individually; a single divergence in relation to the most frequent base within a column was identified in red. Gaps (-) were introduced to optimize the alignment.

**Figure 10 cells-09-02425-f010:**
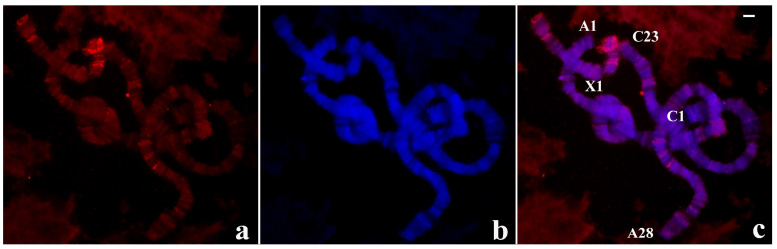
(**a**) Localization of the p*Tp*-2.4 probe (red signal) in chromosomes of *T. pubescens.* (**b**) The same chromosomes stained with DAPI and (**c**) the merged hybridization and DAPI signals. Bar = 15 µm.

**Figure 11 cells-09-02425-f011:**
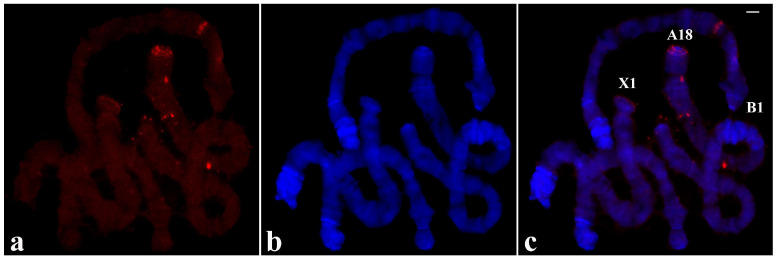
(**a**) Localization of the p*Tp*-2.1 probe (red signal) in chromosomes of *R.americana.* (**b**) The same chromosomes stained with DAPI and (**c**) the merged hybridization and DAPI signals. Chromosome tips identified in the figure show the probe labelling restricted to A-18 and X-1. Bar = 15 µm.

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
