# Peer review of "Chromosome End Diversification in Sciarid Flies"

_cells, 2020, doi:10.3390/cells9112425_

Round 1
Reviewer 1 Report
Generally this manuscript provides an interesting insight on the chromosome end diversification in Sciarid flies. However, there are several questions regarding this manuscript.
- General question for the whole manuscript: For each chromosome, there are two ends: one is telomere and another is centromere. Like in Figure 1: polytene chromosmes of R. Americana, A-1, A-18, B-1,C-1, X-1 are telomeric ends, while B-15, C-11, X-12 and probably A-11 are centromeric ends. Please double check previous chromosome maps of Americana and T. pubescens (Ref 15 and 23) to make sure that you interpreted the maps correctly. Or find more articles to confirm the chromosomal ends of these two species.
- For the chromosome staining, in my experience, Yoyo is better staining agent than DAPI, please read our articles titled “Comparative physical genome mapping of malaria vectors Anopheles sinensis and Anopheles gambiae”(Malaria Journal)) and “Structural divergence of chromosomes between malaria vectors Anopheles lesteri and Anopheles sinensis”(Parasites & Vectors). For DAPI staining, we cannot see clearly the banding pattern of chromosomes, but Yoyo stain can make the bands visible for observers.
- Fig 3-11, please indicate the each chromosomal arm with letters on each figure, so it will be easier to read.
- Figure 3: I think the images tell us that the tip of R. Americana was hybridized to the tip of B-1 with major signal but also yield minor signals on C-1, A-18 and A-1 telomeric ends.
- Figure 5: the conclusion: Use of a probe made of two distinct tips of T. pubescens chromosomes resulted in labelling restricted to two chromosome ends corresponding to microdissected tips used for the probe synthesis (Figure 5), was made based on the Fig 5. I donot think the conclusion stands because (1) no C-23 signal can be seen; (2) There are several signals localized on the ends and interstitial sites. The strongest signal is on A-18 and the signal on B-15 is even stronger than B1. If this result is not correct so it might affect the whole manuscript.
- Line 226-228, add this figure into the manuscript.
Based on the sequence analysis, it seems that B-1 tip and X-1 tip did not contain highly repetitive elements. To rule out the minor signals generated by repetitive elements, you can block the repetitive elements when you perform in situ hybridization.
Author Response
1 General question for the whole manuscript: For each chromosome, there are two ends: one is telomere and another is centromere. Like in Figure 1: polytene chromosmes of R. Americana, A-1, A-18, B-1 ˇC-1, X-1 are telomeric ends, while B-15, C-11, X-12 and probably A-11 are centromeric ends. Please double check previous chromosome maps of Americana and T. pubescens (Ref 15 and 23) to make sure that you interpreted the maps correctly. Or find more articles to confirm the chromosomal ends of these two species.
I thank you for the observation and I know exactly the origin of your doubt that makes complete sense. The problem is not in the references or in the chromosome maps that are correct this was checked and there is no mistake. In fact, here was a problem in Cells office at the time to make the pdf. This will be explained below in the paragraph that you numbered 5 .
2 For the chromosome staining, in my experience, Yoyo is better staining agent than DAPI, please read our articles titled Comparative physical genome mapping of malaria vectors Anopheles sinensis and Anopheles gambiae (Malaria Journal)) and Structural divergence of chromosomes between malaria vectors Anopheles lesteri and Anopheles sinensis (Parasites & Vectors). For DAPI staining, we cannot see clearly the banding pattern of chromosomes, but Yoyo stain can make the bands visible for observers.
Thank you very much for your suggestion, I do not have Yoyo but it is not difficult to purchase. I know and like very much those articles that are important for the knowledge of Culicidae and usually show superb cytogenetics. However, sometimes it is not possible to rear sciarid larvae that provide good polytene chromosomes so that staining techniques are compromised even using good dyes. And this happened in this work. In addition, these species have been difficult to collect as permanent laboratory cultures are not possible.
3 Fig 3-11, please indicate the each chromosomal arm with letters on each figure, so it will be easier to read.
This was done in the present version.
4 Figure 3: I think the images tell us that the tip of R. Americana was hybridized to the tip of B-1 with major signal but also yield minor signals on C-1, A-18 and A-1 telomeric ends.
This usually occurs because the microdissected ends always carry subtelomeric DNA that results in more intense labelling at those tips from which the probe was made. But even so, the results in other non centromeric tips of this species are of significant intensity if you look at panel a. Differences in intensity also appeared when cloned probes were used indicating that the repeat amount per chromosome end may vary.
5 Figure 5: the conclusion: Use of a probe made of two distinct tips of T. pubescens chromosomes resulted in labelling restricted to two chromosome ends corresponding to microdissected tips used for the probe synthesis (Figure 5), was made based on the Fig 5. I donot think the conclusion stands because (1) no C-23 signal can be seen; (2) There are several signals localized on the ends and interstitial sites. The strongest signal is on A-18 and the signal on B-15 is even stronger than B1. If this result is not correct so it might affect the whole manuscript.
You are completely right and I will explain the origin of the problem. When the manuscript was submitted, the managing editor asked me to resend figures 5 and 6 as I did not send a pdf file, something wrong happened with the Cells template file. Now I see, the figures are misplaced when the pdf was built. This was corrected in the present version.
6 Line 226-228, add this figure into the manuscript. Based on the sequence analysis, it seems that B-1 tip and X-1 tip did not contain highly repetitive elements. To rule out the minor signals generated by repetitive elements, you can block the repetitive elements when you perform in situ hybridization.
This figure you asked me to add appears as Supplementary Figure S1.
If I did not miss the point, I think you refer to B-1 tip of R. americana. It does have repeats but they have not been exactly the focus of this manuscript. X-1 does not seem to contain highly repetitive DNA. Your suggestion has not been done because we wanted to be hybridization results representing repetitive elements. It is interesting that, even unblocking repeats, results in T. pubescens point the absence of highly repetitive DNA in chromosome ends.
Finally, I would like to thank you very much for the work you have done, your interest, corrections and suggestions directed to the improvement of the manuscript
Reviewer 2 Report
Review for Cells—Chromosome end Diversification in Sciarid Flies
In the manuscript “Chromosome End Diversification in Sciarid Flies” the author presents data showing that chromosome ends in species from two different genera of flies from the Sciaridae have divergent sequences. While the authors state that chromosome ends are conserved (usually) within a genus, they present evidence that they are different among genera. They reach this conclusion through probes and plasmid microlibraries, in situ hybridization, cloning and sequencing.
Generally, I feel this is an interesting paper and topic, but I feel this manuscript needs some heavy editing to sell the story to a more general audience. In general, I find the results and discussion can be tightened up to removed redundancies between the two sections and to make the story as clear as possible. Otherwise, I have two big points/questions/suggestions that I feel should be done to make the paper more understandable and to help present the importance of this work within the larger literature.
Firstly, I feel there needs to be a bit more information on the evolutionary relationships of these two species: how long have they been diverged, is there a common chromosome end structure you would expect to give weight to how different T. pubescens is, what is the phylogenetic structure of this group? Furthermore, (as the authors state in line 481), chromosome end diversification may occur in distinct genera within a dipteran family. But, I think this really needs to be discussed further/made clearer throughout the introduction. This point is kind of messy and it is never super clear what you mean by this diversity and whether it is commonly this diverse? and is it stable within genera and not between/among genera? Etc.
One big question I have is related to the author’s closing statement on “number of dipterans that have been studied with a focus on chromosome ends” not representing the organism diversity within this order. And while this point is true, much of this work has been done in three families (chironomidae, drosophilidae, and culicidae (possibly)). Do you think this is a strong representation of the entirety of Diptera? I think it is a bit skewed, and therefore not really surprising that you find different results within Sciaridae. I think you should discuss this "skew" in chromosome data for Diptera and highlight how the results you have demonstrate that much of the knowledge of diversity of chromosome ends in Diptera is lacking due to this skew. Can you further discuss this point?
The manuscript also has a large number of grammatical issues that need to be resolved. Many sentences are very long and have a lot of qualifying statements that make them difficult to understand.
Other specific comments:
Lines 8-9: Confusing sentence in terms of structure. Are you saying that Diptera exhibit a remarkable diversity of chromosome end structures even though short terminal repeats synthesized by telomerase are not present.
Lines 10-11: Are you talking about a specific family of Diptera, or are you referring to any given family? And even then, you have a further qualifier of "the same genus". Do you mean: "Within dipteran families, structure of chromosome termini is conserved within genera"?
Lines 16-17: I'm not entirely sure what you are trying to say by this sentence, at least on a first read through.
Lines 18-19: Are you trying to say that "While both species share homology in terminal sequences, these sequences are underrepresented in T. pubescens"?
Lines 20-21: strange phrasing "under this focus". maybe "...compared to other dipterans investigated".
Lines 27-28: grammar is off. Widespread and broadly conserved are redundant. Maybe: Telomeric structure composed of short tandem repeats synthesized by telomerase is broadly conserved across taxa.
Lines 28-29: "However, in Diptera, which display unexpected structural diversity in chromosome ends, telomerase is not present."
Lines 29-31: should be cleared up, as D. virilis is within Drosophila, yet you say Drosophila have specific retrotransposons, then say D. virilis has retrotransposons and terminal satellites?
Line 35: chromosome ends
Lines 38-41: A bit lost here. Needs clarification
Line 65: needs country info
Line 66: what about relatively humidity and light:dark cycle information?
Line 75: I didn't realize you were looking at chrionomus species until here?? Can you describe more in intro/results/discussion what these results mean and why you are doing them
Line 149: "enables comparisons of how evolutionary change shaped..."
Line 159: should be “also has”
Figure 4: There are more labels on 3C, can you add labels like that to 4C?
Line 200: In regards to “what is usually expected”-- which are, what? What is expected?
Line 215: can you give a phylogeny or some other reference to how distantly related they are?
Lines 230-232: wording is a bit strange. Could be rephrased
Line 251-253: newly sterilized material? What does "separated by long time intervals" mean? A specific amount of time?
Lines 279-282. This is written as if it hasn't been done. (it’s in future tense) Has it been completed? If so, it should be past tense
Line 287: should be “to”? not “o”
Line 289: What do you mean by significant similarity? Is there a quantitative measure that you use?
Line 306: should be “represented”
Line 315: databases is one word
Line 361-363. First sentence is long and could be reordered and shortened to make your point. Also, order does not need to be capitalized
Line 372-373: can you give a reference for this statement?
Line 376-377: needs to be reworded. Confusing as is
Line 377-379: needs to be reworded. Confusing as is
Line 385: I think it should be “extents”
Author Response
Generally, I feel this is an interesting paper and topic, but I feel this manuscript needs some heavy editing to sell the story to a more general audience.
Your hard work has helped to present a much better version of the manuscript. Thank you very much and sorry for taking trouble over you..
† In general, I find the results and discussion can be tightened up to removed redundancies between the two sections and to make the story as clear as possible.
Discussion was shortened in the new manuscript version in an attempt to reduce redundancies.
Otherwise, I have two big points/questions/suggestions that I feel should be done to make the paper more understandable and to help present the importance of this work within the larger literature. Firstly, I feel there needs to be a bit more information on the evolutionary relationships of these two species:† how long have they been diverged, is there a common chromosome end structure you would expect to give weight to how different T. pubescens is, what is the phylogenetic structure of this group?
You refer to the same point when you discuss line 215. Information in the literature is very limited but now it is cited in the Introduction. It also appears after your comments on line 215.
Furthermore, (as the authors state in line 481), chromosome end diversification may occur in distinct genera within a dipteran family. But, I think this really needs to be discussed further/made clearer throughout the introduction. This point is kind of messy and it is never super clear what you mean by this diversity and whether it is commonly this diverse? and is it stable within genera and not between/among genera?
I agree with you, it has not been clear the meaning of diversity. Sequence divergence among dipteran repeats or transposons is the first basic sign of diversity as telomeric sequences are not the same even within genera. But in terms of structure, it is possible to to visualize three main groups. In the new version of manuscript, examples of structural diversity were given at the end of the third paragraph of the Introduction, I think they illustrate the concept in dipteran telomeres. Please find these informations in the first (1) and fourth (2) paragraphs in Introduction as well as below:
1-Telomeric structure composed of short tandem repeats synthesized by telomerase is broadly conserved across taxa. However, telomerase is not present in Diptera [1] that displays unexpected diversity in relation to chromosome ends. Telomeric sequences vary even between species of the same genus and provide an example of telomere diversification. Also, dipteran telomeric structures diverge depending on DNA repeat types and how they are organized at the very end of the chromosomes.
2- M-22, M-16 and T-14 short tandem repeats represent a divergent chromosome end structure compared to long (complex) terminal repeats that have been characterized in chironomids and in Anopheles. D. melanogaster constitutes a third divergent structure composed of specific mobile elements in place of tandem repeats.
Etc. † One big question I have is related to the author s closing statement on number of dipterans that have been studied with a focus on chromosome ends not representing the organism diversity within this order. And while this point is true, much of this work has been done in three families (chironomidae, drosophilidae, and culicidae (possibly)).† Do you think this is a strong representation of the entirety of Diptera?† I think it is a bit skewed, and therefore not really surprising that you find different results within Sciaridae.†
We are in agreement with the low representation of Diptera in telomere studies, this was stated at the end of Conclusion in the first version of the manuscript to answer your question.
Honestly, novel results in T. pubescens were not expected before starting the the work. We thought that, like Rhynchosciara, short tandem repeats would be identified by in situ hybridization in the eight non centromeric ends of this species (Tp chromosomes do not have centromeric ends). This feature could be interesting to reinforce that centromeric/pericentric ends of R. americana and chironomid species evolved to have sequences other than those found in non centromeric ends. This was the only motivation to study T. pubescens. However the first results that emerged were unexpected and difficult to interpret, so that we were about to give up studying this fly. Then, the idea of exploiting diversity came to us. Now you see a dataset that could be considered as "expected" by exploiting different genera but it seems expected just because it is the first study comparing results from two genera in the same family.
I think you should discuss this "skew" in chromosome data for Diptera and highlight how the results you have demonstrate that much of the knowledge of diversity of chromosome ends in Diptera is lacking due to this skew.† Can you further discuss this point?
In the last paragraph of Conclusion, a brief discussion on this point was added and reproduced below. I do not know if it is to your satisfaction:
By exploiting a single sciarid species, T. pubescens, this investigation indicates that chromosome end diversification may occur in distinct genera within a dipteran family. Flies that have been studied with a focus on chromosome ends are far from representing the organism diversity within this order. To broaden this scenario, basic studies are necessary in order to keep other dipterans in the laboratory, a task that is not always possible. Also, cytogenetics that is needed in telomere research cannot be performed in several species. However, there are genera still unexploited within Drosophilidae, Chironomidae, Culicidae, Psychodidae and Sciaridae that can be raised in the laboratory and also allow chromosome research. The introduction of new dipteran genera in chromosome end studies may show structures other than those that have already been characterized to date.
The manuscript also has a large number of grammatical issues that need to be resolved. Many sentences are very long and have a lot of qualifying statements that make them difficult to understand.
I am really sorry for taking trouble over you with my limitation in expressing some ideas. Thank you very much for your patience and help.
Other specific comments: Lines 8-9:† Confusing sentence in terms of structure. Are you saying that Diptera exhibit a remarkable diversity of chromosome end structures even though short terminal repeats synthesized by telomerase are not present.
The sentence was modified as follows: Dipterans exhibit a remarkable diversity of chromosome end structures in contrast to the conserved system defined by telomerase and short repeats.
† Lines 10-11: Are you talking about a specific family of Diptera, or are you referring to any given family? And even then, you have a further qualifier of "the same genus".† Do you mean:† "Within dipteran families, structure of chromosome termini is conserved within genera"?
The sentence was replaced by the above one you suggested.
† Lines 16-17:† I'm not entirely sure what you are trying to say by this sentence, at least on a first read through.
The sentence refers to two possible interpretations for the results observed in T. pubescens that appear also in Discussion. It was slightly modified in an attempt to make it more clear: The data argue for the existence of either specific terminal DNA sequences for each chromosome tip in T. pubescens, or sequences common to all chromosome ends but their extension does not allow detection by in situ hybridization.
† Lines 18-19:† Are you trying to say that "While both species share homology in terminal sequences, these sequences are underrepresented in T. pubescens"?
Yes, but perhaps homology is not exactly the term. In any case, the sentence was replaced by another one very close to your suggestion: Both sciarid species share terminal sequences that are significantly underrepresented in chromosome ends of T. pubescens.
† Lines 20-21:† strange phrasing "under this focus".† maybe "...compared to other dipterans investigated".
The sentence was modified exactly according to your suggestion.
† Lines 27-28:† grammar is off. Widespread and broadly conserved are redundant.† Maybe:† Telomeric structure composed of short tandem repeats synthesized by telomerase is broadly conserved across taxa.
The sentence was modified exactly according to your suggestion.
† Lines 28-29:† "However, in Diptera, which display unexpected structural diversity in chromosome ends, telomerase is not present."
The sentence was modified exactly according to your suggestion.
† Lines 29-31:† should be cleared up, as D. virilis is within Drosophila, yet you say Drosophila have specific retrotransposons, then say D. virilis has retrotransposons and terminal satellites?
Although it seems strange, that information was based on a talk with James Mason who worked for a long time in Drosophila telomeres. Complex satellite-like repeats were first characterized in this species and they are present in all telomeres of D. virilis. Eventually, another paper described retrotransposons in this species but if you see the paper, they are localized in a few chromosome termini. So the question still remains whether repeats or retrotransposons compose the very end of the virilis. Anyway, the sentence was modified as follows: In D. virilis, complex terminal satellites in addition to retrotransposons were identified at chromosome ends [5, 6].
† Line 35: chromosome ends
Corrected as you suggested.
† Lines 38-41: A bit lost here.† Needs clarification
The text was partially modified to become more clear: Complex (414bp) tandem repeats were first characterized in this sciarid fly that were found to be sub-telomeric rather than truly terminal repeats [20]. A second repeat type named M-22 is unusually short for a repeat tandemly arrayed at the chromosome ends of dipterans. Additional data showed that M-22 tandem repeats lie distal to sub-telomeric repeat arrays [21].
† Line 65:† needs country info Country added for the two species Line 66:† what about relatively humidity and light:dark cycle information?
light:dark cycle information was included. Larvae are kept in boxes contain humid soil and food but relative humidity has never been measured and for this reason it has been omitted not only in this manuscript but also in previous ones.
† Line 75: I didn't realize you were looking at chrionomus species until here?? Can you describe more in intro/results/discussion what these results mean and why you are doing them
Chironomus riparius was in fact used as a control to microdissection procedures. In Results (sub-topic Controls), use of this species was justified in detail. Please have a look, I think it is possible to understand. In the Introduction, chironomid telomeres appear as one important example of structural diversity of chromosome ends in dipterans. In Discussion, the first paragraph describes the importance of methods employed in this work and the reference used shows that Chironomus pioneered research in dipteran telomeres even before Drosophila.
Line 149:† "enables comparisons of how evolutionary change shaped..."
Sentence modified exactly according to your suggestion
† Line 159:† should be also has Corrected
Figure 4:† There are more labels on 3C, can you add labels like that to 4C?
New versions of Figs. 4, 5, 7 and 10 have more labels to identify polytene chromosomes.
Line 200:† In regards to what is usually expected -- which are, what? What is expected?
This sentence was deleted and replaced by the following: In this sense, R. americana data agree with chironomid terminal repeats that were found to be specific to non-centromeric ends.
† Line 215:† can you give a phylogeny or some other reference to how distantly related they are?
Sciarid phylogeny has long been a complicate matter and for this reason information was omitted in the first manuscript version. References now appear in the Introduction and below you have the informations included that, to be frank, do not help very much:
Relationships within the family Sciaridae have long been controversial. In an early report, Trichomegalosphys and Rhynchosciara appear in distinct tribes, Megalosphyni and Sciarini respectively [24]. Still considering early classifications, another study placed the two genera in different groups, I and II [25]. More recently, molecular phylogeny as well as cladograms on sciarid relationships have been produced [26] but analyses including T. pubescens and R. americana have not been available yet. Despite the lack of molecular data, genera placed in distinct tribes or even groups strongly suggest that they are significantly divergent.
Lines 230-232:† wording is a bit strange.† Could be rephrased
The sentence was rephrased: The results described above showed that T. pubescens and R. americana share sequences at chromosome ends as well as in other genomic regions despite divergent labelling patterns in chromosomes of the two species.
† Line 251-253:† newly sterilized material? What does "separated by long time intervals" mean? A specific amount of time?
The sentence was modified specifying the time interval between microdissections of the two species: Microdissection procedures involving chromosomes of R. americana and T. pubescens were separated by a five year interval in order to discard the possibility of contamination and always performed with newly sterilized material.
† Lines 279-282. This is written as if it hasn't been done.† (it s in future tense) Has it been completed? If so, it should be past tense
According to your correction: Total genomic DNA labelled for hybridization is enriched with highly repetitive DNA. Therefore, it would be able to produce, by comparison to single copy or even middle repetitive DNA, significant hybridization signals in plasmid inserts representing highly repetitive sequences.
Line 287:† should be to ? not o
Thanks, already corrected.
† Line 289:† What do you mean by significant similarity? Is there a quantitative measure that you use?
In general >60% base matches comprising at least 50% of the query extension.
Line 306:† should be represented
Thanks, corrected in the present manuscript version.
Line 315: databases is one word
Thanks, corrected in the present manuscript version.
Line 361-363. First sentence is long and could be reordered and shortened to make your point. Also, order does not need to be capitalized
According to your observations: Given the diversity of sequences composing chromosome termini in Diptera, massive sequencing methods cannot help to provide information on chromosome ends of species in this order.
Line 372-373: can you give a reference for this statement?
Chironomid references that appear in the Introduction were given at the end of the sentence.
Line 376-377: needs to be reworded.† Confusing as is † Line 377-379:† needs to be reworded. Confusing as is
The new version for these lines:
The data allow to suggest two possibilities for the structure of chromosome termini of this species. The first possibility could be seen as quite unusual and implies specific sequences for each chromosome tip. The second, more conservative possibility argue for the existence of DNA sequences common to all chromosome ends in T. pubescens. However, limits possibly imposed by the short extension of these sequences impede detection in chromosome ends by in situ hybridization.
Line 385:† I think it should be extents
Yes, it is corrected in the present manuscript version †
Reviewer 3 Report
Dear Dr. Gorab,
I have carefully read your manuscript entitled "Chromosome end diversification in sciarid flies". I believe this is an important contribution to our knowledge of telomere diversity in the family Sciaridae as well as in the order Diptera in general. However, I would like to suggest a few minor corrections to the text. First, I recommend to refer to a recently published review of telomere structure in insects by Kuznetsova et al. (https://doi.org/10.1111/jzs.12332) in the Introduction. Second, all taxonomic categories must include only lowercase letters, e.g. "suborder" instead of "Sub-Order" (line 32). Moreover, older genus/species names should be only briefly cited at the first appearance of the newer ones, e.g. "Trichomegalosphys (= Trichosia) pubescens" (please also change "Trichosia" to "Trichomegalosphys" in line 62). In addition, please remove a redundant full stop at the end of the sentence in line 111. Please also change "o" and "a single a chromosome tip" to "to" and "a single chromosome tip" (lines 287 and 367 respectively).
Yours sincerely,
Author Response
Dear Dr.,
I thank you very much for carefully reading the manuscript and for your comments and suggestions directed to the manuscript improvement.
I am really sorry for the omission. The reference suggested was added in the Introduction [4] as well in the reference list.
Sub-Order was replaced by suborder on line 32.
Genera names were corrected according to your observations
Double full stop you found in Materials and Methods was deleted
Mistakes that appeared in lines 287/367 were already corrected.
Round 2
Reviewer 1 Report
I am satisfied with the revised manuscript,and all comments were responded soundly and properly.
Author Response
Dear Reviewer,
Once again, thank you very much for carefully reading the manuscript and for helping to improve it with your comments, observations and suggestions as well.
With best regards
Eduardo
Reviewer 2 Report
Upon reviewing the manuscript “Chromosome end diversification in sciarid flies” for a second time, I find the manuscript greatly improved. I think the points are much clearer and the manuscript as a whole sells the story much better. Overall, the paper is a good read. Well done. Nice improvements. I'm happy to see the work out.
I have a few minor points that should be addressed before publication. They are quick and should be easy to address.
Line 28: put comma after Diptera and then say "an order which displays unexpected diversity in relation to chromosome ends".
Line 33-34: Should have commas surrounding “in addition to retrotransposons”
Line 45: "of chromosome ends in R. americana"
Line 88-90: I would state that "were prepared as a control as described above.....". This will make the use of Chironomus riparius as a control for the study much more clear.
Line 129-131: This sentence needs to be clarified. There is use of the word "either", yet I only see one "option" in the sentence. I would expect an "either/or" statement. Is there a second option here? Or is either not the correct word.
Line 234: comma after “Flies”
Line 264: should be "use of the R. americana", or "use of a R. americana" or "DNA probes". This will fix the grammatical issue in the sentence.
Line 484: Should be “distantly related”, I think.
Line 493: typo: should be “remains” not “remais”
Line 499-500: Maybe, "An additional issue that persists is that cytogenetic methods necessary for telomore research cannot always be performed in several species." I had to read this a couple times to get the point.
Author Response
I have no words to thank you for your enormous effort directed to the improvement of the manuscript. Your observations were all included in the manuscript.
With kind regards
Eduardo